# Huntingtin phosphorylation governs BDNF homeostasis and improves the phenotype of *Mecp2* knockout mice

Yann Ehinger[1,†,¶], Julie Bruyère[2,¶], Nicolas Panayotis[1,‡], Yah-Se Abada[3], Emilie Borloz[1], Valérie Matagne[1], Chiara Scaramuzzino[2], Hélène Vitet[2], Benoit Delatour[3], Lydia Saidi[1,§], Laurent Villard[1], Frédéric Saudou[2,*] (iD) & Jean-Christophe Roux[1,**] (iD)

## Abstract

**Mutations in the X-linked *MECP2* gene are responsible for Rett syndrome (RTT), a severe neurological disorder for which there is no treatment. Several studies have linked the loss of MeCP2 function to alterations of brain-derived neurotrophic factor (BDNF) levels, but non-specific overexpression of BDNF only partially improves the phenotype of *Mecp2*-deficient mice. We and others have previously shown that huntingtin (HTT) scaffolds molecular motor complexes, transports BDNF-containing vesicles, and is under-expressed in *Mecp2* knockout brains. Here, we demonstrate that promoting HTT phosphorylation at Ser421, either by a phospho-mimetic mutation or inhibition of the phosphatase calcineurin, restores endogenous BDNF axonal transport *in vitro* in the corticostriatal pathway, increases striatal BDNF availability and synaptic connectivity *in vivo*, and improves the phenotype and the survival of *Mecp2* knockout mice—even though treatments were initiated only after the mice had already developed symptoms. Stimulation of endogenous cellular pathways may thus be a promising approach for the treatment of RTT patients.**

**Keywords** Mecp2; Rett; BDNF; huntingtin; axonal transport
**Subject Categories** Genetics, Gene Therapy & Genetic Disease; Neuroscience; Pharmacology & Drug Discovery

## Introduction

MeCP2 (Methyl-CpG-binding protein 2) is one of the most abundant proteins in the brain, yet the precise nature of its activities remains controversial. It was originally discovered as a DNA methylation-dependent transcriptional repressor (Meehan *et al*, 1992), but has also been shown to play various roles in chromatin compaction, global gene expression, alternative splicing, and miRNA processing (Young *et al*, 2005; Chahrour *et al*, 2008; Skene *et al*, 2010; Cheng *et al*, 2014). Interest in this protein rose sharply after it was discovered that mutations in *MECP2* cause Rett syndrome (RTT), a severe developmental disorder (Amir *et al*, 1999; Lyst & Bird, 2015). Females with RTT begin life apparently healthy, but between 6 and 18 months of age they undergo regression of early milestones, with deterioration of motor skills, eye contact, speech, and motor control; they then develop a range of neurological symptoms, including anxiety, respiratory dysrhythmias, and seizures (Lyst & Bird, 2015; Katz *et al*, 2016). Whereas loss of MeCP2 function leads to RTT, duplication or triplication of the locus leads to intellectual disability, autistic features, and motor dysfunction, as observed in males with *MECP2* duplication syndrome (Van Esch, 2012).

Disease-causing mutations in *MECP2* alter the expression of thousands of genes (Chahrour *et al*, 2008). Among these, brain-derived neurotrophic factor (BDNF), a neuronal modulator that plays a key role in neuronal survival, development, and plasticity (Cheng *et al*, 2011), is one of the best studied (Chen *et al*, 2003, 2015; Chang *et al*, 2006; Sampathkumar *et al*, 2016). BDNF appears to be involved in the appearance and progression of the RTT phenotype in mice: *Mecp2* knockout mice (KO) present lower BDNF levels, and conditional BDNF deletion in *Mecp2* KO mice accelerates the onset of RTT-like symptoms (Chang *et al*, 2006). Conversely, conditional BDNF overexpression in the brain of *Mecp2* knockout mice leads to an improvement of certain locomotor and electrophysiological deficits (Chang *et al*, 2006). Although it is possible that the incomplete rescue reflects that MeCP2 has much broader regulatory effects than BDNF, it is possible that BDNF overexpression fails to restore

1  Aix Marseille Univ, INSERM, MMG, UMR_S 1251, Marseille, France
2  Univ. Grenoble Alpes, Inserm, U1216, CHU Grenoble Alpes, Grenoble Institut Neurosciences, GIN, Grenoble, France
3  Sorbonne Université, Institut du Cerveau et de la Moelle épinière, ICM, Inserm U1127, CNRS UMR 7225, Paris, France
   *Corresponding author. Tel: +33 4565 20514; E-mail: frederic.saudou@inserm.fr
   **Corresponding author. Tel: +33 4913 24904; E-mail: jean-christophe.roux@univ-amu.fr
   ¶These authors contributed equally to this work
   †Present address: Department of Neurology, University of California, San Francisco, San Francisco, CA, USA
   ‡Present address: Department of Biomolecular Sciences, Weizmann Institute of Science, Rehovot, Israel
   §Present address: Department of Psychiatry and Neuroscience, Faculty of medicine, Centre de recherche CERVO, Université Laval, Quebec City, QC, Canada

synaptic and neuronal function because it does not target the appropriate neurons. In support of this hypothesis, recent evidence suggests that MeCP2 deficiency leads to the disruption of cell-autonomous and autocrine BDNF signaling in excitatory glutamatergic neurons, and that increasing BDNF levels in diseased neurons restores their growth and ability to form synapses (Sampathkumar *et al*, 2016).

We have previously shown that BDNF homeostasis and transport involve huntingtin (HTT) and HTT-associated protein 1 (Hap1) (Roux *et al*, 2012; Saudou & Humbert, 2016). Levels of both these proteins are lower than normal in excitatory cortical neurons deficient for *Mecp2* (Roux *et al*, 2012). Here, we tested whether activating HTT, by a genetic or pharmacological approach, could improve BDNF homeostasis in *Mecp2*-deficient neuronal circuits and in *Mecp2* KO mice.

## Results

### BDNF transport is slowed in Mecp2-deficient axons

To examine BDNF transport in Mecp2-deficient neurons, we took advantage of the recent development and validation of microfluidic devices to reconstitute a neuronal network and monitor intracellular dynamics (Taylor *et al*, 2010; Virlogeux *et al*, 2018). We focused here on the corticostriatal network, which is altered in RTT and *Mecp2* KO mice (Roux *et al*, 2012; Xu *et al*, 2014). The device consists of a presynaptic compartment containing cortical neurons, a postsynaptic chamber containing striatal neurons, and a synaptic chamber containing corticostriatal contacts. Cortical neurons plated in the first compartment extend axons that connect to striatal dendrites within the synaptic compartment (Fig 1A), thus creating an oriented corticostriatal network on-a-chip (Virlogeux *et al*, 2018). This network reproduces the physiological network as we previously showed that cortical neurons from embryonic day (E) 15.5 are enriched in CTIP2/TBR1 neurons that correspond to the deepest layers of the cortex that send axons to the striatum. Also, most striatal neurons correspond to enkephalin-positive neurons that are the output-projecting neurons of the striatum (Virlogeux *et al*, 2018). Cortical neurons were transduced with mCherry-tagged BDNF lentivirus (BDNF-mCherry), and we used spinning disk confocal videomicroscopy to record the dynamics of BDNF-mCherry containing vesicles within microchannels in the axons (Fig 1A and Movie EV1). The dynamics of vesicles are in agreement with our recent studies using this technology (Moutaux *et al*, 2018; Virlogeux *et al*, 2018).

We transfected neurons with *Mecp2* siRNA, which reduced Mecp2 protein levels by 63% compared to WT (Fig EV1A). This reduced the mean velocity and overall linear flow of BDNF vesicles reaching the corticostriatal contacts (Fig 1B). The number of moving vesicles was unchanged (Fig EV1B).

### HTT phosphorylation status influences anterograde and retrograde BDNF transport

Overexpression of phosphorylated HTT at S421 leads to increased outward transport in cells (Colin *et al*, 2008), but the role of endogenous HTT phosphorylation in axons remains unknown. We

isolated cortical neurons from homozygous knock-in mice in which S421 of HTT was replaced by an alanine ($HTT^{S421A/S421A}$ or $HTT_{SA}$), mimicking the absence of phosphorylation, or by an aspartic acid ($HTT^{S421D/S421D}$ or $HTT_{SD}$), mimicking constitutive phosphorylation (Thion *et al*, 2015). Mutating HTT at S421 abrogated the capacity of our phospho-HTT specific antibody to recognize endogenous phosphorylated protein (Fig EV1C). Importantly, the S into A or into D mutations had no effect on HTT protein levels, as quantification showed no difference in HTT protein expression between WT, $HTT_{SD}$, and $HTT_{SA}$ mice (Fig EV1C). Also, we found that the mutations had no effect on cell viability, as shown by the MTT assay (Fig 1C). We then reproduced a corticostriatal network by plating the neurons within microfluidic devices. Phospho-mimetic HTT ($HTT_{SD}$) significantly increased the speed of BDNF vesicles moving in the anterograde direction (Fig 1D, Movie EV1) without affecting the number of moving BDNF vesicles (Fig EV1D). In contrast, the absence of HTT phosphorylation ($HTT_{SA}$) significantly increased the retrograde velocity of BDNF vesicles (Fig 1D). Thus, HTT phosphorylation status influences the velocity of BDNF vesicles in axons.

We next investigated whether HTT phosphorylation status influences the observed dysregulation of BDNF transport in *Mecp2* siRNA-transfected neurons (Figs 1E and EV1E and F). Phospho-mimetic HTT ($HTT_{SD}$) rescued both anterograde and retrograde transport of BDNF, along with the mean velocity of BDNF vesicles and linear flow (Fig 1E). Preventing HTT phosphorylation ($HTT_{SA}$) restored only the retrograde velocity of BDNF and linear flow rate to control levels (Fig 1E). The overall effect of HTT phosphorylation on BDNF transport under normal or low-Mecp2 conditions was not due to a change in the number of moving BDNF vesicles (Fig EV1D and F) or in cell viability, since we observed no toxicity in *Mecp2* siRNA-transfected $HTT_{SD}$ or $HTT_{SA}$ neurons compared to *Mecp2* siRNA-transfected WT neurons (Fig 1F). These results demonstrate that genetically promoting HTT phosphorylation at S421 rescues the transport of BDNF vesicles in projecting corticostriatal siMecp2 neurons.

### Constitutive phosphorylation of HTT rescues corticostriatal BDNF transport and increases postsynaptic TrkB phosphorylation and markers of postsynaptic density *in vivo*

*Mecp2* KO mice show altered corticostriatal connections and reduced BDNF levels in the striatum (Roux *et al*, 2012). We therefore assessed the impact of huntingtin phosphorylation on BDNF transport *in vivo*. We crossed *Mecp2* KO mice with either $HTT_{SA}$ or $HTT_{SD}$ mice. The resulting double-mutant male mice, deficient for the *Mecp2* gene (KO) and homozygous for the S421A ($HTT^{S421A/S421A}$) or S421D mutation ($HTT^{S421D/S421D}$), will from hereon be referred to as KO/$HTT_{SA}$ and KO/$HTT_{SD}$, respectively. Most of the BDNF protein located in the striatum comes from the cortex by anterograde transport within corticostriatal afferences (Altar *et al*, 1997). We therefore quantified the level of BDNF proteins using ELISA at the site of translation (the cortex) and the target site (the striatum) of 55-day-old WT mice, KO, KO/$HTT_{SD}$, and KO/$HTT_{SA}$ mice (Fig 2A). The ratio of striatal and cortical BDNF is an indicator of the *in vivo* efficacy of BDNF axonal transport to the corticostriatal synapses. The ratio we observed in KO/$HTT_{SD}$ mice ($1.57 \pm 0.3$) was equivalent to what we observed in

WT mice (1.57 ± 0.6) and was significantly higher than that in *Mecp2* KO mice (1.14 ± 0.1) or KO/HTT$_{SA}$ mice (1.1 ± 0.2). These results suggest that HTT phosphorylation rescues corticostriatal BDNF transport *in vivo*.

We also found that improved corticostriatal BDNF transport in KO/HTT$_{SD}$ mice increased levels of TrkB phosphorylation at the postsynaptic level compared to KO WT (+36%, $P < 0.05$) and

KO/HTT$_{SA}$ (+35%, $P < 0.05$), showing that the BDNF release is stimulated *in vivo* (Figs 2A and EV1G). As a consequence, the post-synaptic marker PSD-95 is increased in KO/HTT$_{SD}$ striatum compared to KO/WT striatum (+37%, $P < 0.01$) (Figs 2A and EV1G). We conclude that promoting HTT phosphorylation stimulates striatal BDNF pathways and helps maintain corticostriatal synapse homeostasis *in vivo*.

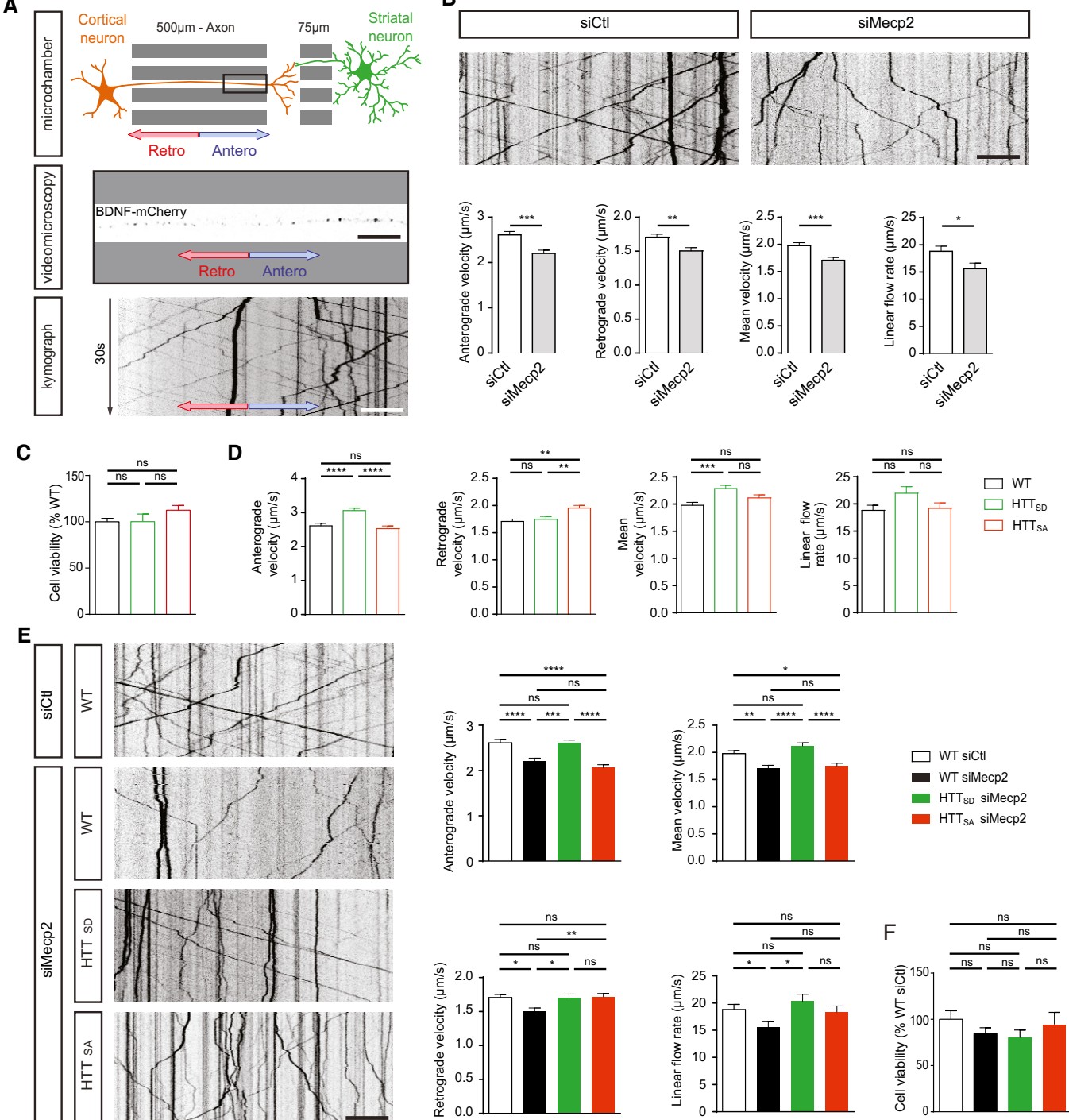

Figure 1.

**Figure 1.  Huntingtin phosphorylation rescues BDNF transport in Mecp2-deficient axons.**

A   Microchamber that allows the isolation of axons within a corticostriatal network and live-cell imaging of axonal BDNF-mCherry transduced into mouse primary cortical neurons.

B   Representative kymographs and quantification of anterograde, retrograde, mean velocity, and linear flow rate of BDNF-mCherry-containing vesicles in cortical neurons transfected with siMecp2 ($n$ = 753 vesicles/81 axons) or siControl (siCtl) ($n$ = 894 vesicles/94 axons), inducing significant silencing of *Mecp2* (3 independent experiments; unpaired *t*-test).

C   Quantification of anterograde, retrograde, mean velocity, and linear flow rate of BDNF-mCherry trafficking into cortical neurons obtained from WT ($n$ = 753 vesicles/ 81 axons), $HTT_{SA}$ ($n$ = 812 vesicles/81 axons), or $HTT_{SD}$ ($n$ = 787 vesicles/82 axons) homozygous knock-in mice in which S421 of HTT was replaced by an alanine ($HTT^{S421A/S421A}$ or $HTT_{SA}$), mimicking the absence of phosphorylation, or by an aspartic acid ($HTT^{S421D/S421D}$ or $HTT_{SD}$), mimicking constitutive phosphorylation (3 independent experiments; one-way ANOVA test with Tukey's multiple comparisons).

D   Cell viability in WT ($n$ = 3), $HTT_{SA}$ ($n$ = 1), or $HTT_{SD}$ ($n$ = 2) neurons measured by MTT assay. No differences were observed between the different groups (three independent experiments; one-way ANOVA Kruskal–Wallis test, $P$ = 0.5701)

E   Kymographs and quantification of anterograde, retrograde, mean velocity, and linear flow rate of BDNF-mCherry axonal trafficking into WT ($n$ = 894 vesicles/94 axons), $HTT_{SA}$ ($n$ = 812 vesicles/82 axons), or $HTT_{SD}$ ($n$ = 787 vesicles/94 axons) cortical neurons transfected with siCtl or siMecp2, which significantly silenced Mecp2 (three independent experiments; one-way ANOVA with Tukey's multiple comparisons).

F   Cell viability in WT, $HTT_{SA}$, or $HTT_{SD}$ neurons transfected with siCtl or siMecp2 measured by MTT assay (WT siCtl: $n$ = 3; WT siMecp2: $n$ = 3; $HTT_{SD}$ siMecp2: $n$ = 3; $HTT_{SA}$ siMecp2: $n$ = 2). No differences were observed between the different groups (three independent experiments; one-way ANOVA Kruskal–Wallis test, $P$ = 0.5701).

Data information: Scale bars = 20 μm. $*P < 0.05$, $**P < 0.01$, $***P < 0.001$, $****P < 0.0001$, ns = not significant. Data are presented as the means ± SEM.

## Constitutive phosphorylation of HTT reduces the loss of body weight of *Mecp2* KO mice and extends their lifespan

We next investigated whether manipulating HTT phosphorylation *in vivo* could have an effect *on Mecp2* KO mouse symptoms. We first assessed the behavior of $HTT_{SA}$ and $HTT_{SD}$ homozygous mice using a modified SHIRPA primary screen (Appendix Table S1) and various behavioral assays (Fig EV2A–D) and found no significant differences between WT and $HTT_{SA}$ or $HTT_{SD}$ mice at 6 months in motor activity, strength, coordination, exploratory behavior, or body weight. *Mecp2* KO mice carrying the S421D mutation (KO/ $HTT_{SD}$) had a longer lifespan than *Mecp2* KO mice, though they still were subject to premature lethality, whereas KO/$HTT_{SA}$ mice showed no improvement over that of *Mecp2* KO mice (Fig 2B). KO/ $HTT_{SD}$ mice also had greater body weight than the *Mecp2* KO mice, whereas the absence of phosphorylation in the KO/$HTT_{SA}$ mice had no effect on weight (Figs 2C and EV2E).

## Constitutive phosphorylation of HTT reduces apneas in *Mecp2* KO mice

Breathing disturbances are prominent and deleterious in RTT patients (Kerr *et al*, 1997) and *Mecp2* KO mice (Viemari *et al*, 2005). We found that apnea frequency increased with age in *Mecp2* KO mice, from P35 to P55 (Fig 2D). The KO/$HTT_{SD}$ mice had significantly fewer apneas than *Mecp2* KO mice at both time points (Fig 2D). The absence of HTT phosphorylation slightly worsened this phenotype.

These data suggest that promoting HTT phosphorylation *in vivo* improves respiration in *Mecp2* KO mice.

## Constitutive phosphorylation of HTT improves motor function of *Mecp2* KO mice

We next investigated motor coordination on the accelerating rotarod test (Pratte *et al*, 2011). In agreement with previous studies, *Mecp2* KO mice showed a progressive, significant decrease in the latency to fall relative to WT mice (Fig 2E). Promoting HTT phosphorylation delayed the appearance of motor incoordination until P55, but ablating HTT phosphorylation worsened motor coordination at both time

points (Fig 2E). HTT phosphorylation had no effect on the overall exploration pattern of *Mecp2* KO mice in the open-field (OF) test (Fig EV2F), but the time before initiation of the first movement was significantly longer for the *Mecp2* KO than WT mice (Fig 2F). The latency to explore the OF arena was further increased in KO/$HTT_{SA}$ mice, whereas it was reduced in the KO/$HTT_{SD}$ mice toward the values observed for WT mice (Fig 2F).

We next monitored circadian activity and found no deregulation in the circadian rhythm in the different genotypes during the day (Fig EV2G). There was, however, a striking suppression of spontaneous locomotion in the absence of *Mecp2* during the dark phase (7:00 PM–7:00 AM) (Fig EV2G). Promoting HTT phosphorylation enhanced spontaneous night activity, increasing the distance travelled by the KO/$HTT_{SD}$ mice, although it did not reach the value of the WT mice (Fig EV2H). Conversely, the distance travelled by the KO/$HTT_{SA}$ mice was significantly less than that covered by the KO/ $HTT_{SD}$ mice. Overall, these results show that promoting HTT phosphorylation improves sensorimotor coordination and locomotor activity in *Mecp2* KO mice.

## Inhibition of calcineurin by FK506 restores BDNF transport in Mecp2-silenced axons

Since chronic phosphorylation rescues BDNF trafficking *in vitro* and improves symptoms in mice, we investigated whether pharmacological induction of HTT phosphorylation could be of therapeutic value in RTT. We previously reported that HTT phosphorylation can be inhibited by the protein phosphatase calcineurin (Pardo *et al*, 2006; Pineda *et al*, 2009). We therefore evaluated whether FK506, a calcineurin inhibitor, can restore BDNF transport in *Mecp2*-silenced neurons by increasing HTT phosphorylation. Cortical neurons connected to striatal neurons within the microfluidic devices were transduced with the BDNF-mCherry lentiviral vector and a siRNA targeting either a control sequence or that of *Mecp2*. Five days after plating, we incubated the microfluidic chambers for 1 h with 1 μM FK506 or vehicle and recorded BDNF axonal transport.

FK506 treatment increased HTT phosphorylation (Fig 3A) and rescued the reduced BDNF trafficking measured after Mecp2 silencing (Fig 3B and C, Movie EV2). Both mean anterograde and

retrograde vesicle velocities were increased in si*Mecp2*-transfected neurons, with a significant overall effect on mean velocity and linear flow (Fig 3C). The number of moving vesicles did not change (Fig 3C). FK506-induced calcineurin inhibition thus mitigates deficits of BDNF transport observed in *Mecp2*-silenced neurons.

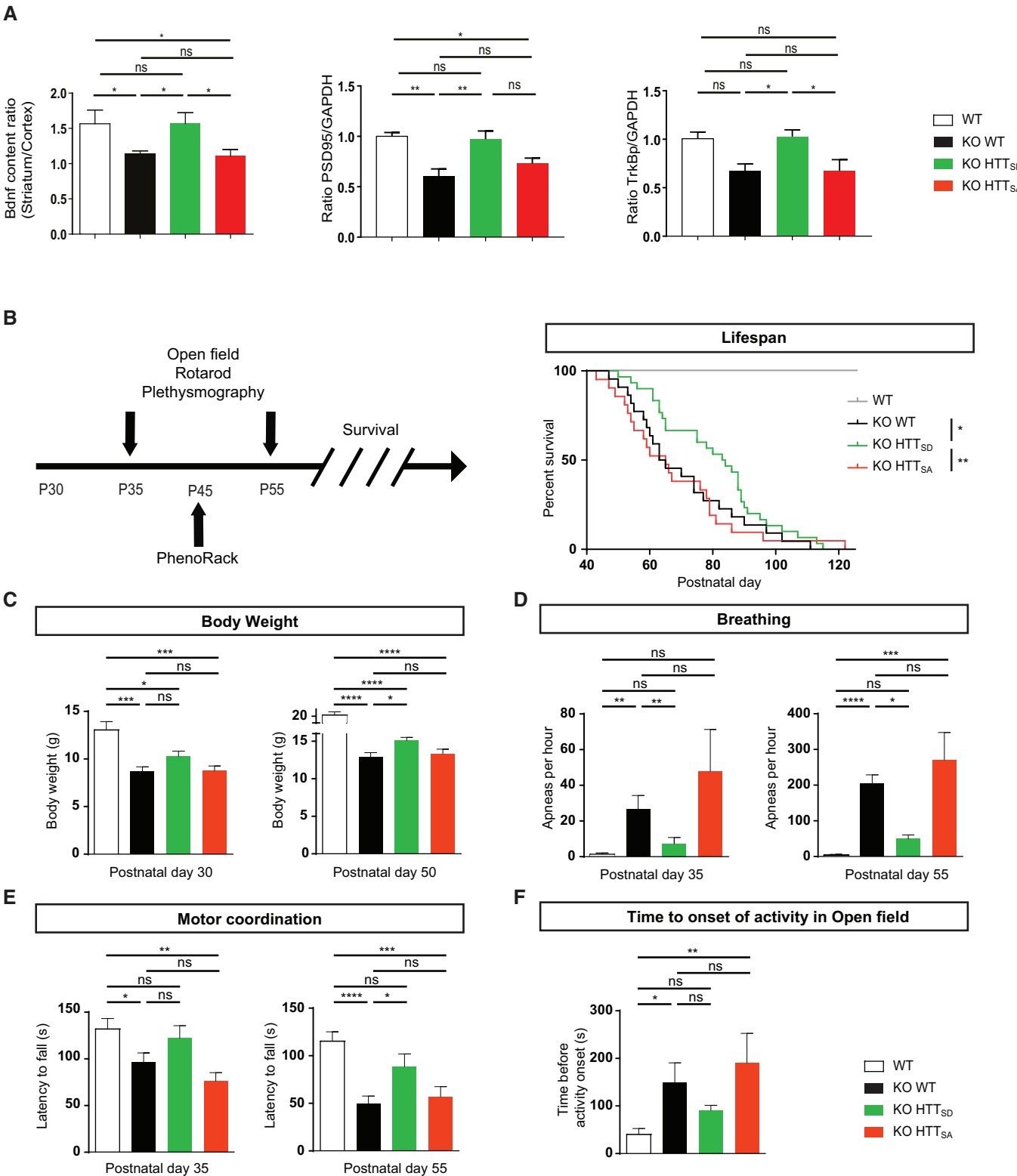

**Figure 2.**

◄

**Figure 2.  The S421D mutation improves BDNF corticostriatal transport *in vivo*, the motor and autonomic functions of *Mecp2*-deficient mice and increases their lifespan.**

A  Left: The ratio between the striatal and the cortical BDNF determined by quantifying the level of BDNF proteins at the cortex and the striatum of 55-day-old WT ($n = 8$), *Mecp2* KO ($n = 6$), *Mecp2* KO/HTT$_{SD}$ (mimicking the absence of phosphorylation) ($n = 4$), and *Mecp2* KO/HTT$_{SA}$ mice (mimicking constitutive phosphorylation) ($n = 5$). Since striatal BDNF depends only on BDNF transport from the cortex, this ratio reflects BDNF transport through corticostriatal pathway (Mann–Whitney test). Right: Quantitative analysis of phospho-TrkB protein level in striatum of KO WT mice and KO HTT$_{SD}$ mice by immunoblotting. The relative expression levels of phospho-TrkB were normalized against GAPDH and are presented as the ratio ($n = 6$ mice per group) (Mann–Whitney test). Middle: Quantitative analysis of PSD-95 protein level in striatum of *Mecp2* KO/HTT WT mice and KO/HTT$_{SD}$ mice by immunoblotting. The relative expression levels of PSD-95 were normalized against GAPDH and are presented as the ratio ($n = 6$ mice per group) (Mann–Whitney test).

B  Left: We investigated the behavior of *Mecp2* KO mice at 35, 45, and 55 days of age and assessed their survival. Right: KO/HTT$_{SD}$ mice ($n = 30$) and WT mice ($n = 17$) had a significantly longer lifespan than KO ($n = 22$) or KO/HTT$_{SA}$ mice ($n = 21$) (Kaplan–Meier survival test).

C  Body weight of 10 WT, 24 KO, 32 KO/HTT$_{SD}$, and 24 KO/HTT$_{SA}$ mice at P30 and P50 (one-way ANOVA with Tukey's comparison).

D  Frequency of apnea of 9 WT, 10 KO, 14 KO/HTT$_{SD}$, and 9 KO/HTT$_{SA}$ mice at P35 and P55 (Kruskal–Wallis test with Dunn's comparison).

E  KO/HTT$_{SD}$ mice ($n = 19$) performed as well as WT ($n = 17$) at P35 on the accelerating rotarod test and continued to outperform *Mecp2* KO ($n = 16$) and KO/HTT$_{SA}$ ($n = 13$) at P55 (one-way ANOVA with Fisher's LSD test).

F  Time before the onset of spontaneous locomotor activity of 17 WT, 20 KO, 19 KO/HTT$_{SD}$, and 17 KO/HTT$_{SA}$ mice at P55. (Kruskal–Wallis test with Dunn's comparison).

Data information: Data are presented as the means $\pm$ SEM. *$P < 0.05$, **$P < 0.01$, ***$P < 0.001$, ****$P < 0.0001$, ns = not significant.

## FK506 increases HTT phosphorylation in *Mecp2* KO mice

To determine whether calcineurin inhibition via FK506 treatment could improve *Mecp2* KO mouse symptoms, we treated *Mecp2* KO, WT and HTT$_{SA}$ mice with FK506 to induce HTT phosphorylation. Mice were first injected intraperitoneally with FK506 (5 mg/kg) and sacrificed 2 h after administration, as previously described (Pardo *et al*, 2006). We analyzed HTT phosphorylation at S421 by immunoblotting fresh whole-brain protein extract (Figs 4A and EV3A and B). As previously described, administration of FK506 in WT mice increased HTT phosphorylation, by about 1.3-fold (Pardo *et al*, 2006). In *Mecp2* KO mice, FK506 doubled HTT phosphorylation relative to vehicle. FK506 had no effect on HTT phosphorylation in HTT$_{SA}$ mice (Fig EV3A and B).

## FK506 treatment improves respiration and motor function of *Mecp2* KO mice and extends their lifespan via HTT phosphorylation

To determine whether FK506 could improve the *Mecp2*-deficient phenotype, we treated *Mecp2* KO mice by intraperitoneal FK506 injection (10 mg/kg), three times a week, starting at P30. We choose P30 as a starting point, since the onset of the *Mecp2*-knockout phenotype is already apparent at this stage as breathing abnormalities, locomotor deficits, and body weight loss (Guy *et al*, 2001; Viemari *et al*, 2005; Matagne *et al*, 2017). FK506 treatment significantly increased the lifespan of *Mecp2* KO mice (Fig 4B) and

induced a significant increase in body weight by P55 (Fig 4C). The number of apneas was significantly lower in the FK506-treated group than the vehicle group at P35 and remained lower at P55 (Fig 4D). FK506-treated *Mecp2* KO mice performed better on the accelerating rotarod at P50 (Fig 4E) and showed better forelimb muscle strength than the vehicle group (Fig 4F). We next investigated cell death in the striatum of these mice. In accord with previous reports (Reiss *et al*, 1993; Kishi & Macklis, 2004; Armstrong, 2005; Roux *et al*, 2007), we did not detect any cell death in *Mecp2* KO mice by cleaved caspase-3 immunodetection or by TUNEL assay (Fig EV4A and B). Importantly, we found that FK506 treatment did not induce any neuronal toxicity or cell death after 20 days of treatment (Fig EV4A and B).

We also verified the induction of HTT phosphorylation in *Mecp2*-deficient mice after 20 days of FK506 chronic treatment. Mouse brain samples were analyzed 2 hrs after the last treatment. We did not detect any difference in Mecp2 protein levels or in HTT phosphorylation in wild-type mice treated with FK506 (Fig EV3B and C). However, at this time point we did not detect elevated HTT phosphorylation in the samples treated with FK506 relative to DMSO-treated animals, likely because repeated treatments desensitize the pathway. We observed a trend toward reduction in cFos staining in *Mecp2*-deficient mice that was increased back to control levels in FK506-treated *Mecp2* KO mice, suggesting a positive treatment effect (Fig EV4C and D).

To establish that the beneficial effect of FK506 is mediated by HTT phosphorylation at S421, we treated KO/HTT$_{SA}$ mice by intraperitoneal FK506 injection as previously done (Fig 4).

**Figure 3.  FK506 restores BDNF axonal transport in *Mecp2*-silenced axons.**

A  Western blot analysis of DIV 5 cortical neurons transfected with siMecp2 or siControl (siCtl) and treated with 1 μM FK506 or vehicle for 1 h, and quantification of pS421 HTT ($n = 9$ per group). Data are presented as means $\pm$ SEM, *$P < 0.05$, **$P < 0.01$, ***$P < 0.001$, Mann–Whitney test.

B  Representative kymographs showing axonal trafficking of BDNF-mCherry-containing vesicles in cortical neurons transfected with siMecp2 or siControl (siCtl) and treated with 1 μM FK506 or vehicle for 1 h. Scale bar = 20 μm.

C  Quantification of anterograde, retrograde, mean velocity, linear flow rate, and number of BDNF-mCherry-containing vesicles from the data in (A) (siMecp2+Vehicle: $n = 801$ vesicles/95 axons; siMecp2+FK506: $n = 1020$ vesicles/116 axons; siCtl+Vehicle: $n = 780$ vesicles/83 axons; siCtl+FK506: $n = 1,029$ vesicles/102 axons). Data are presented as the mean $\pm$ SEM of at least three independent experiments (one-way ANOVA with Tukey's comparison). *$P < 0.05$, **$P < 0.01$, ***$P < 0.001$, ****$P < 0.0001$, ns = not significant.

Source data are available online for this figure.

▶

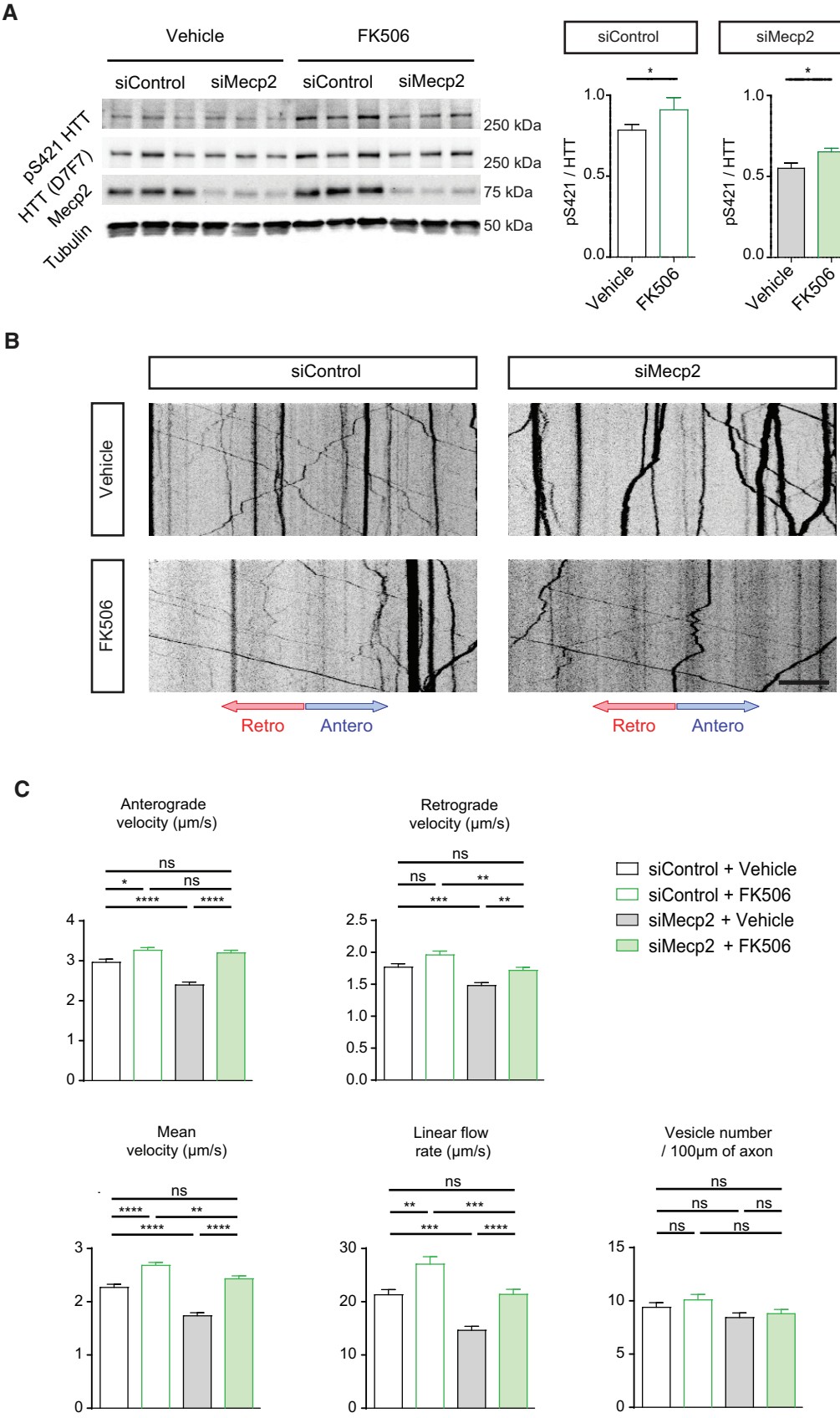

**Figure 3.**

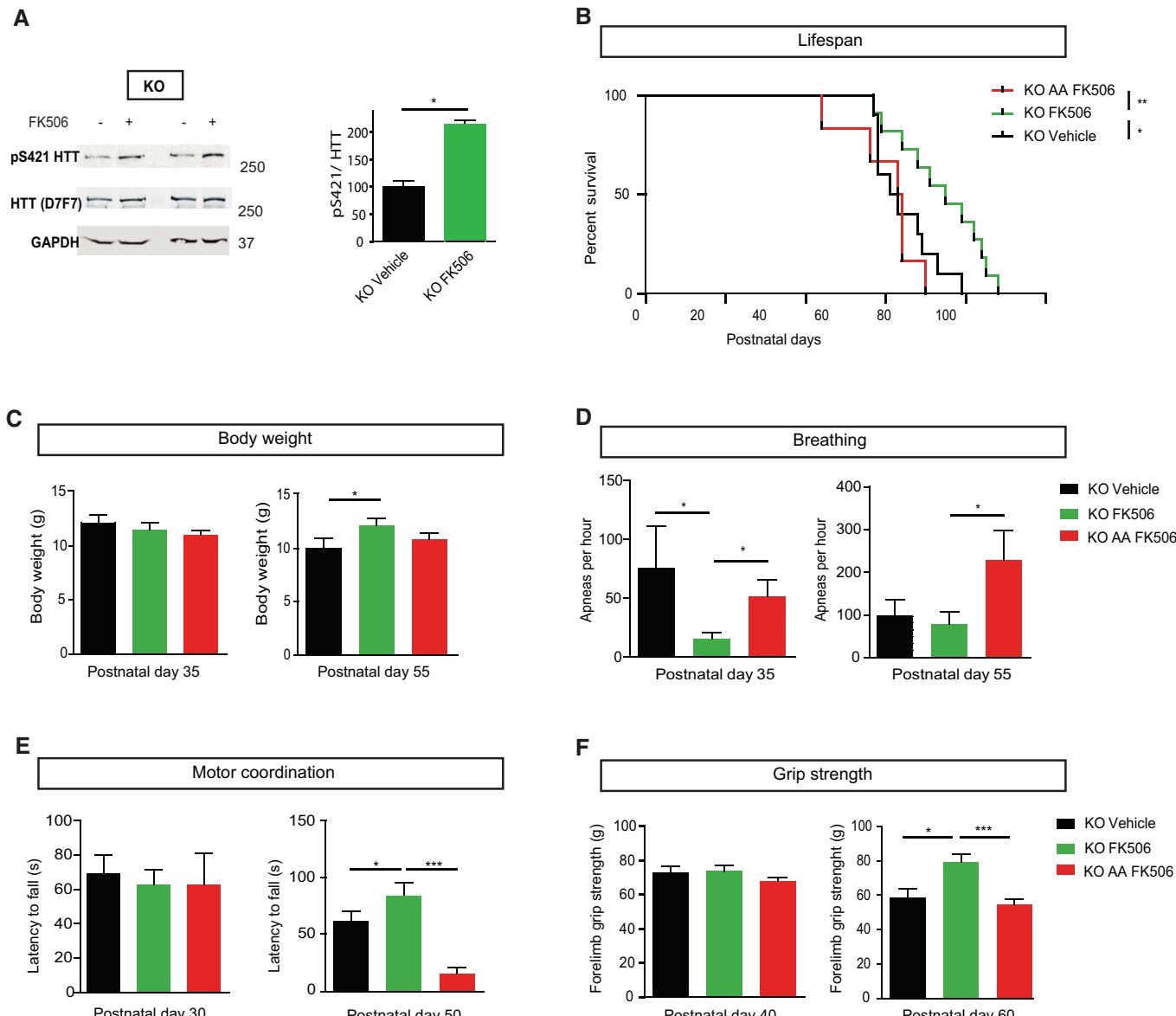

**Figure 4. Calcineurin inhibition by FK506 in *Mecp2*-deficient mice improves motor and autonomic functions and extends lifespan through huntingtin-dependent phosphorylation.**

A  Western blot of HTT S421 phosphorylation. We treated 30-day-old *Mecp2* KO mice intraperitoneally with FK506 (5 mg/kg) or vehicle and analyzed brain extracts for endogenous HTT phosphorylation by Western blotting, 2 h after administration, using an anti-phospho-HTT-S421 specific antibody. The D7F7 antibody recognizes total HTT. The relative protein level of phospho-HTT was normalized on total HTT protein level and is presented as the ratio (KO FK506 $n = 4$, KO Vehicle $n = 4$). Data are presented as means ± SEM, *$P < 0.05$, Mann–Whitney test).

B–F  We treated 30-day-old *Mecp2* KO mice ($n = 10$) and KO/HTT$_{SA}$ mice ($n = 10$) with 5 mg/kg FK506 three times a week by intraperitoneal injection and assessed them in various behavioral tests. (B) FK506-treated KO mice lived longer than vehicle-treated KO mice and FK506-treated KO HTT$_{SA}$ mice (Kaplan–Meier survival test). (C) Body weight of FK506-treated KO, FK506-treated KO HTT$_{SA}$, and vehicle-treated KO mice at P35 and P55 (Mann–Whitney test). (D) Frequency of apnea of FK506-treated KO, FK506-treated KO HTT$_{SA}$, and vehicle-treated KO mice at P35 and P55 (Mann–Whitney test). (E) Motor coordination of FK506-treated KO, FK506-treated KO HTT$_{SA}$, and vehicle-treated KO mice on the accelerating rotarod test at P30 and P50 (Mann–Whitney test). (F) Forelimb strength of FK506-treated KO, FK506-treated KO HTT$_{SA}$, and vehicle-treated KO mice assessed by the grip strength test at P40 and P60 (Mann–Whitney test).

Data information: *$P < 0.05$, **$P < 0.01$, ***$P < 0.001$, ns = not significant. Data are means ± SEM.
Source data are available online for this figure.

Interestingly, the lifespan of *Mecp2* KO/HTT$_{SA}$ mice was significantly shorter than that of *Mecp2* KO mice (Fig 4B). Moreover, FK506 treatment affected neither motor function nor apneas of

KO HTT$_{SA}$ mice (Fig 4D–F). These results demonstrate that HTT phosphorylation at S421 is essential for the therapeutic effect of FK506.

## Discussion

Here, we demonstrate that genetically or pharmacologically inducing HTT phosphorylation at S421 rescues BDNF vesicular transport in *Mecp2*-silenced projecting corticostriatal neurons and improves several pathophysiological features of *Mecp2* KO mice. The ability of FK506 as a proof-of-principle candidate to improve Mecp2 symptoms highlights the feasibility of pharmacological stimulation of HTT S421-P in a mouse model of RTT. Importantly, our findings show that HTT phosphorylation can stimulate endogenous machinery to promote BDNF trafficking in the corticostriatal network.

There has been considerable interest in modulating BDNF expression and signaling as a treatment for RTT. Unfortunately, BDNF itself has very low blood–brain barrier (BBB) permeability, precluding its peripheral administration as a potential therapy. Several studies have used indirect stimulation of BDNF metabolism via fingolimod (Deogracias *et al*, 2012) or ampakine treatment (Ogier *et al*, 2007) to circumvent this limitation, but these pharmacological treatments only partially improved the phenotype of *Mecp2* KO mice. Daily injection of IGF-1, another neurotrophic factor that can cross the BBB and is known to induce Akt phosphorylation (Humbert *et al*, 2002), has been found to improve survival, locomotor activity, and respiratory rhythm in *Mecp2* KO mice (Tropea *et al*, 2009). Finally, potential agonists of the TrkB receptor, such as 7,8-dihydroxyflavone (Johnson *et al*, 2012) and LM22A-4 (Schmid *et al*, 2012; Kron *et al*, 2014; Li *et al*, 2017), improved breathing patterns in *Mecp2* KO mice. It is important to note, however, that the improvements observed in these different studies required that pharmacological treatments be initiated before the appearance of the first symptoms.

Given that RTT is not diagnosed until long after symptoms have begun, we searched for a more practical translational approach based on observed improvements in the RTT phenotype by constitutive phosphorylation of HTT. We selected FK506 because Pardo *et al* (2006) found that this drug increases phosphorylation of mutant HTT. Importantly, FK506 also induces HTT phosphorylation in *Mecp2* KO mice. Our results showed that FK506 treatment, starting at an already-symptomatic stage, improved the lifespan, motor strength and coordination, and exploratory behavior, and reduced the frequency of apneas in *Mecp2* KO mice in a manner that requires HTT phosphorylation at S421. Thus, part of the beneficial effect of FK506 treatment is due to the stimulation of the HTT-dependent transport of BDNF. It is possible that FK506 also modulates the trafficking of other cargo, such as mitochondria, which could contribute to the *in vivo* improvement we saw in the *Mecp2* KO mice. Future studies investigating such additional effects would be of interest, as they would increase the therapeutic relevance of FK506 treatment for Mecp2 symptoms (Reddy *et al*, 2012).

We conclude that BDNF trafficking and supply are diminished in the absence of Mecp2 and can be effectively stimulated by promoting HTT phosphorylation. Our results are in accord with a recent study that demonstrated that BDNF acts cell-autonomously in an autocrine loop, as wild-type neurons were unable to rescue growth deficits of neighboring *Mecp2*-deficient neurons (Sampathkumar *et al*, 2016). Thus, HTT phosphorylation may increase the bioavailability of BDNF at the synapse through autocrine and paracrine mechanisms in the brains of *Mecp2 KO* mice and likely represents a strategy of therapeutic interest versus a general, non-synapse-specific increase of BDNF levels in the brain.

## Materials and Methods

### Mouse breeding and genotyping

All mouse lines were on a C57BL/6J genetic background. The Mecp2tm1-1Bird *Mecp2*-deficient mice were obtained from the Jackson Laboratory and maintained on a C57BL/6 background by using C56BL/6J male breeders (also from the Jackson Laboratory). The HTT knock-in mice were previously generated by inserting a point mutation in exon 9 (AGC>GAC, Ser>Asp called S421D or AGC>GCC, Ser>Ala called S421A) (Thion *et al*, 2015). To obtain double-mutant mice (*Mecp2* KO and S421A or S421D), heterozygous females ($Mecp2^{+/-}$) were crossed with homozygous S421D or S421A males.

Hemizygous mutant males ($Mecp2^{-/y}$ also called *Mecp2* KO) were generated by crossing heterozygous females ($Mecp2^{+/-}$) with C57BL/6 males. Genotyping was performed by routine PCR technique following a previously described protocol (Roux *et al*, 2012). Animals were housed under standard conditions of temperature ($21 \pm 2°C$) and humidity ($55 \pm 5\%$), with food and water *ad libitum* in a 12:12 h day/night cycle.

Only male mice were used for all genotypes and experiments.

Experimental protocols were approved by the ethical committee of the Aix Marseille University and the French M.E.N.E.S.R. minister (Permit Number: 02910.02).

The experimental procedures were carried out in keeping with the European guidelines for the care and use of laboratory animals (EU directive 2010/63/EU), the guide for the care and use of the laboratory animals of the French national institute for science and health (INSERM). All experiments were made to minimize animal suffering. In order to reduce animal suffering, endpoints were fixed as weight loss limit (below 80% of maximum weight), obvious breathing defects, or severe injury. In these cases, animals were euthanized with an overdose of pentobarbital (100 mg/kg BW i.p., Ceva Santé Animale, La Ballastiere, France).

### FK506 chronic *in vivo* treatment of *Mecp2*-deficient mice (KO and KO/HTT_SA)

*Mecp2* KO mice were randomly assigned in groups. From P30, animals received an i.p. injection three times a week of either 10 mg/kg FK506 in 17% DMSO or vehicle alone (17% DMSO). The first group was used for behavioral testing and evaluating survival (see below). A second group was used to assess the chronic effects of FK506 treatment from P30 to P50 on cellular and molecular parameters.

### Behavioral testing

All mice were weighed every 5 days and assessed for survival. At P35, P45, and P55, each animal underwent a set of behavioral tests to assess motor function, activity, and breathing pattern (Fig 2). All testing occurred during the light phase of the light–dark cycle (except for the PhenoRack monitoring). All the behavioral experiments were performed blinded to HTT genotype and treatment.

### Open field

An arena made of clear perspex (38 × 30 cm), under controlled light conditions (300 lx), was used for the FK506 study. For the genetic study, mice were placed in a 1-m-diameter arena under controlled light conditions (300 lx). Activity was recorded using the Videotrack software (Viewpoint, Lyon, France) for 20 min. Activity and average velocity (cm/s) were determined from the total distance moved and activity duration. Vertical activity (rearing, leaning, and grooming) was noted by an experimenter blind of the genotype.

### Rotarod

Sensorimotor abilities were assessed by the accelerating Rotarod apparatus (Panlab LE-8200, Harvard Apparatus). Each trial started at 4 rpm and reached 40 rpm speed after 300 s. Mice underwent three trials, with 5-min rest time in between. The trial ended when the mouse fell off the rod or after 300 s. Latency to fall (in second) was measured, and only, the best trial was recorded. In the case of mice clinging to the rod, the trial was stopped and the passive rotation was considered a failure in performance like falling (Brown *et al*, 2005).

### Grip strength

A Bioseb grip strength meter (Panlab) was used to measure forelimb strength. Five measures for each mouse were taken, and means were calculated from the three best trials.

### Whole-body plethysmography

To assess apneas, mice were placed in a clear plexiglass chamber (200 ml) and allowed to breathe naturally under conscious and unrestrained conditions. After a ~30-min adaptation, breathing was recorded: The spirogram was obtained by recording the pressure difference between the two chambers, and then, the signal was amplified, filtered and fed to an analog-to-digital converter (sampling frequency, 1 kHz), and finally analyzed by the Spike2 interface and software (v.5.04, Cambridge Electronic Design Ltd., Cambridge, UK). Apneas were defined by more than 1s without breathing, as previously described (Roux *et al*, 2007). Breathing cycles were divided into four groups according to their duration: hyperventilation (including cycles in the range 0–0.3 s range); ventilation (0.3–0.7 s); hypoventilation (0.7–1 s); and apneas (1–∞ s). The breathing variability was calculated as the mean standard variability. Breathing parameters were obtained from the analysis of quiet period of at least 100 consecutive cycles (Fig EV5).

### Home cage activity

To analyze spontaneous activity and circadian rhythm, mice were put in individual cages and monitored by the PhenoRack system (Viewpoint, Lyon, France). Mice locomotion was tracked by infrared light during 48 h, and after a 24-h adaptation phase, only the last 24-h activity was analyzed. The recorded data allowed us to analyze activity, distance (cm), and velocity (cm/sec).

### BDNF immunoassay

P55 ($n = 4$ Wt; $n = 4$ *Mecp2* KO; $n = 4$ KO/HTT$_{SD}$; $n = 5$ KO/HTT$_{SA}$) male mice were euthanized by cervical dislocation, and their brains were dissected out within the first 2 min post-mortem. The cortex and the striatum were microdissected using a punching needle (0.5 mm in diameter). Briefly, brain area dissection was performed on cryostat brain sections with the help of a 5× magnifying lens, following their stereotaxic coordinates (Paxinos and Franklin, 2001). We dissected cortical and striatal samples only coming from the same slice in the same rostro-caudal level. Tissue samples were freshly isolated and lysed in 200 μl of the extraction buffer (100 mM Tris-pH 7.5, 125 mM NaCl, 0.1 mM EGTA, 0.1% Triton X-100, Roche® protease inhibitors cocktail), sonicated, centrifugated. The supernatant was stored at −80°C until assay. Total protein concentration was determined by using the bicinchoninic acid (BCA) protein assay (Thermo Fisher scientific) and measured with a spectrophotometer (Glomax, Promega). The level of BDNF protein from tissue extracts was determined with the BDNF Emax® ImmunoAssay System (Promega) using the manufacturer's instruction. In the present study, we measured only free mature BDNF, and therefore, we proceeded directly to the ELISA protocol avoiding any acid treatment. In each assay, duplicate wells were assigned for each sample. A Victor 4 PerkinElmer microplate reader was used to measure signal intensity from the wells at 450 nm. A linear standard curve was generated with standard BDNF from 5.8 to 500 pg/ml. The total amount of BDNF per well was calculated based on the standard curve, and each sample value was within the linear range. The relative BDNF value was then calculated by normalizing the amount of BDNF against the total amount of protein input.

### Western blotting

Adult (P55) male mice were euthanized by cervical dislocation, and their brains were dissected out within the first 2 min post-mortem. Brains were dissected on ice, and proteins were extracted by sonication and isolated in a lysis buffer containing 50 mM Tris–HCl (pH = 7.5), 150 mM NaCl, 2 mM EGTA, 2 mM EDTA, 1% Triton X-100, 10 nM betaglycerophosphate, 5 mM sodium pyrophosphate, 50 mM sodium fluoride, and Halt$^{TM}$ proteases and phosphatases inhibitor cocktail (Pierce Thermo Fisher). Proteins were extracted from neuronal culture with lysis buffer containing 4 mM HEPES, pH 7.4, 320 mM sucrose, and protease inhibitor cocktail (Roche). Protein concentrations were determined by the bicinchoninic acid method. After a denaturation step at 96°C for 5 min, proteins (100 μg) were separated on 4–20% SDS-polyacrylamide gel (Life technology) and transferred onto a nitrocellulose membrane by electroblotting using the Trans-Blot turbo transfer system (Bio-Rad). The membrane was blocked with blocking buffer (Millipore, WBAVDFL01) for 1 h at room temperature. Primary antibodies for HTT [1:1,000, rabbit, clone D7F7, CST (Lunkes *et al*, 2002)], HTT-pS421 [1:500. rabbit homemade previously described (Humbert *et al*, 2002)], Mecp2 (1:1,000, rabbit, CST), TRKB-p816 (Millipore ABN1381), PSD-95 (1:1,000, Neuromab), Tubulin (1:5,000, mouse, Sigma), GAPDH (1:5,000, rabbit, Sigma), Actin (1:10,000, mouse, Millipore), and Calnexin (1:1,000, rabbit, Sigma C4731) were diluted in the same solution and incubated overnight at 4°C. The membrane was incubated using appropriate HRP secondary antibodies (donkey anti-rabbit 711-035-152 and donkey anti-mouse 715-035-150; Jackson Immunoresearch Laboratory and Bio-Rad ChemiDoc XRS System) or fluorescent secondary antibodies (IRDye 800 CW, IRDye 680 RD, LI-COR, and LI-COR Odyssey Imager). Quantitative analyses of signal intensity were performed using ImageJ software. For

quantification, total (phosphoindependent) protein signals were used to normalize the phospho-protein signal.

## Immunohistochemistry

Mice were euthanized with pentobarbital and transcardially perfused with 0.01 M PBS followed by 4% paraformaldehyde (PFA) in phosphate buffer. Brains were removed, fixed in 4% PFA overnight at 4°C, and then cryopreserved in 20% sucrose for 3 days. Brains were then rapidly frozen and coronally sectioned into 50-μm sections using a Leica VT1200s cryostat (Leica Biosystems). Sections were rehydrated in PBS three times for 10 min and blocked with 7% normal goat serum in PBS for 1 h and then incubated overnight at room temperature (RT) with anti-cFos (1/200, rabbit, Abcam ab209794) or anti-cleaved caspase-3 (1/400, rabbit, Cell signaling) antibodies. The sections were then washed three times for 10 min each in PBS followed by incubation for 2 h with the following secondary antibodies: Alexa Fluor 488-labeled goat anti-rabbit at 1/400. After staining, the sections were washed three times, for 10 min each in PBS, and then incubated 10 min in DAPI. After a last 10-min wash, sections were mounted in Immu-mount (Thermo Scientific). The immunolabeled slices were digitized and recorded using an Apotome Axioimager 2 (Carl Zeiss) or a Zeiss Lumar stereomicroscope coupled to Axiocam digital camera (Axiovision 4.4, Carl Zeiss).

## Videomicroscopy

Embryonic (E15.5) neuronal cultures were prepared as previously described (Liot *et al*, 2013). Ganglionic eminences and cortex were dissected, and dissociated cortical neurons were nucleofected with ON-TARGET plus mouse Mecp2 siRNA or Non-targeting siRNA 1 (Dharmacon) according to the protocol of Amaxa Nucleofection (Lonza). Then, neurons were plated into microchambers coated with poly-D-lysine (0.1 mg/ml) in the cortical and synaptic compartment or poly-D-lysine and laminin (10 μg/ml, Sigma) into the striatal compartment and cultured at 37°C in a 5% $CO_2$ incubator for 5 days. After 24 h in culture, cortical neurons were transduced with lentivector coding for BDNF-mCherry into presynaptic neuron chamber for axonal transport analysis as previously described (Virlogeux *et al*, 2018). Acquisitions were done on microgrooves, at the limit of the synaptic compartment, at 5 Hz for 30 s on inverted microscope (Axio Observer, Zeiss) coupled to a spinning disk confocal system (CSU-W1-T3; Yokogawa) connected to an electron-multiplying CCD (charge-coupled device) camera (ProEM+1024, Princeton Instrument) at 37°C and 5% $CO_2$. Quantifications of vesicle velocity, linear flow rate, and vesicle number were done on 100 μm of axon using KymoTool Box ImageJ plug-in as previously described (Zala *et al*, 2013; Virlogeux *et al*, 2018). Vesicle velocity corresponds to segmental anterograde or retrograde velocity. Directional net flux is the anterograde cumulative distance minus the retrograde cumulative distance. Regarding vesicle number, a vesicle is considered anterograde when the distance travelled by one vesicle is more anterograde than retrograde.

## MTT assay

In a corticostriatal network, at DIV 11 or 12, presynaptic chamber was filled with MTT 1/10$^e$ from MTT solution at 5 mg/ml. After a 3.5-h incubation at 37°C, MTT solution was carefully removed from the presynaptic chamber and MTT solvent was added (10% SDS, 1.2 mM HCl) for 30 min under agitation. Then, MTT solvent was removed from the microfluidics device and put in a well within a 96-well plate. The absorbance was read at 490 nm on a microplate reader (PHERAstar FS, BMG labtech). Negative controls (dead cells) were treated with $H_2O_2$ at 100 mM for 48 h before the experiment.

## Terminal deoxynucleotidyl transferase dUTP nick end labeling (TUNEL) of apoptotic cells

Cell apoptosis was measured *in vivo* using the Click-It™ Plus TUNEL Assay (Invitrogen) according to the manufacturer's protocol.

## Statistical analysis

All analyses were performed using GraphPad Prism for Windows/MacOS (GraphPad Software, La Jolla, California, USA, www.graphpad.com). The results are reported as mean ± standard error of the mean (SEM). A $P < 0.05$ was considered to be statistically significant. For group comparisons, normality distribution of the datasets was tested before performing any statistical test by Shapiro's test. In case of non-normal distribution, a non-parametric test was performed. When ANOVA was used, Brown Forsythe's test verifies that the variance is homogenous. One-way ANOVAs were performed as indicated with Dunnett's post-hoc analysis for pairwise comparisons when normal distribution. Kruskal–Wallis test with Dunn's multiple comparison was performed on datasets without normal distribution. When appropriate, two groups were compared with an unpaired two-way *t*-test or Mann–Whitney test on datasets without normal distribution. The Kaplan–Meier log-rank test was used for survival studies.

**Expanded View** for this article is available online.

## Acknowledgements

We thank S. Humbert for sharing the *HTT*$^{S421A}$ and *HTT*$^{S421D}$ mice and for advice; the GIN imaging facility (PIC-GIN) for help with image acquisitions; G. Froment, D. Nègre, and C. Costa from the lentivirus production facility of SFR Biosciences (UMS3444/CNRS, US8/Inserm, ENS de Lyon, UCBL); and V. Brandt for critical reading of the manuscript. This work was supported by grants from Agence Nationale pour la Recherche (ANR-2012-BSV1-0003-03 ANTARES, J.C.R. & F.S., ANR-14-CE35-0027-01 PASSAGE, F.S.; ANR-15-JPWG-0003-05 EU Joint Program-Neurodegenerative Disease (JPND) Research project CircProt (no. 643417), (F.S.), Fondation pour la Recherche Médicale (FRM, équipe labellisée, F.S.), Fondation Bettencourt Schueller (F.S.), Fédération pour la Recherche sur le Cerveau (F.S.), Inserm (J.C.R. & F.S.), Aix-Marseille Université (J.C.R.), AFSR (J.C.R.), Institut d'etablissement d'AMU "Marseille Maladies Rares" (MarMaRa) (J.C.R) Promex Stiftung Für Die Forschung (JCR), Rettsyndrome.org (JCR), and NeuroCoG in the framework of the "Investissements d'avenir" program (ANR-15-IDEX-02, F.S.). F.S. laboratory is member of the Grenoble Center of Excellence in Neurodegeneration (GREEN).

## Author contributions

YE, JB, NP, Y-SA, BD, LV, FS, and JCR designed experiments. YE, JB, NP, Y-SA, LS, VM, CS, EB, and HV performed experiments. YE, JB, NP, Y-SA, LS, VM, CS, HV, BD, LV, FS, EB, and J-CR analyzed the data, and YE, JB, FS, and J-CR wrote the manuscript.

**The paper explained**

**Problem**

Rett syndrome (RTT) is a severe neurodevelopmental disorder that becomes apparent in the second year of life, robbing girls of the motor, language, and social skills they acquired and replacing them with learning impairments, stereotypic behaviors, seizures, respiratory dysrhythmias, and a host of other symptoms. RTT is caused by mutations in the X-linked gene *MECP2* (Methyl-CpG-binding protein 2), which encodes the MeCP2 protein, an epigenetic factor that governs the expression of thousands of neuronal genes. Even if gene therapy becomes possible, the introduction of the *MECP2* gene into a cell already expressing a wild-type copy would cause a problem of MeCP2 dosage (MeCP2 duplication syndrome is an equally severe disorder). We therefore need viable approaches to therapy. Several studies have linked the loss of MeCP2 function to reductions in the levels of BDNF (brain-derived neurotrophic factor) in the brain of RTT patients as well as in mouse models. BDNF is crucial to learning and memory, but transport of BDNF-containing vesicles from the cortex to the striatum is abnormally reduced in Mecp2 knockout (KO) mouse neurons. Intriguingly, BDNF is transported within neurons by the Huntingtin protein (HTT), so named because mutations in HTT underlie Huntington's disease. In the present study, we tested whether promoting HTT's transport function by stimulating its phosphorylation at serine 421 (S421) might restore BDNF transport.

**Results**

We used genetic and pharmacological approaches to promote HTT phosphorylation at S421 in both Mecp2-deficient neurons and *Mecp2* knockout mice. We also evaluated the consequences of HTT S421 phosphorylation on BDNF axonal trafficking in projecting corticostriatal neurons using microfluidic devices that mimic the excitatory network. We found that promoting huntingtin phosphorylation restores the intracellular trafficking of BDNF in *Mecp2*-silenced neurons, dramatically improves several key symptoms in *Mecp2* knockout mice, and extends their lifespan.

**Impact**

Our data demonstrate that promoting BDNF transport in the appropriate neuronal circuits is more effective at restoring normal function in Mecp2-deficient neurons than non-specific BDNF overexpression. Stimulation of endogenous cellular pathways, such as HTT phosphorylation at S421, may provide a promising new approach for the treatment of RTT patients.

## Conflict of interest

The authors declare that they have no conflict of interest.

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
