## [Review Process File · EMBO Molecular Medicine]

Huntingtin phosphorylation governs BDNF homeostasis and improves the phenotype of Mecp2 knockout mice

Yann Ehinger, Julie Bruyère, Nicolas Panayotis, Yah-Se Abada, Emilie Borloz, Valérie Matagne, Chiara Scaramuzzino, H  l  ne Vitet, Benoit Delatour, Lydia Saidi, Laurent Villard, Fr  d  ric Saudou, Jean-Christophe Roux

Review timeline:

Submission date:	16 May 2019
Editorial Decision:	19 June 2019
Revision received:	18 October 2019
Editorial Decision:	14 November 2019
Revision received:	27 November 2019
Accepted:	5 December 2019

Editor: C  line Carret

Transaction Report:

1st Editorial Decision

19 June 2019

Thank you for the submission of your manuscript to EMBO Molecular Medicine. We have now heard back from the three referees whom we asked to evaluate your manuscript.

Overall, the referees find the study novel and of interest. While ref. 1 is mainly concerned about the clinical relevance of the approach, this referee and the other two referees ask for additional controls, details, and experiments to improve the conclusiveness of the findings and we would like to insist on those. As a proof of concept study, we would be happy to invite revision on the condition that experiments, and missing data be provided and add to the findings. We also would encourage you to improve the *in vivo* data performance and analysis along the ref.1's comments. We do however, realise that this could potentially lead to a lot of additional work and time and as such, should you already have additional *in vivo* data, we would like to see it but won't insist on it as a prerequisite for resubmission.

We would therefore welcome the submission of a revised version within three months for further consideration and would like to encourage you to address all the criticisms raised as suggested to improve conclusiveness and clarity. Please note that EMBO Molecular Medicine strongly supports a single round of revision and that, as acceptance or rejection of the manuscript will depend on another round of review, your responses should be as complete as possible.

Please also contact us as soon as possible if similar work is published elsewhere. If other work is published, we may not be able to extend the revision period beyond three months.

I look forward to receiving your revised manuscript.

***** Reviewer's comments *****

Referee #1 (Remarks for Author):

Rett syndrome (RTT) is the first cause of severe intellectual disability in girls and it is generally caused by mutations in MECP2. Although studies on mouse models of this disorder prove that RTT is reversible, at least in mice, no effective cure is yet available. Mecp2 deficiency leads to the deregulation of several genes; among these the down regulation of BDNF is generally accepted as having a role in the disorder. Accordingly, several pre-clinical and clinical trials aim at fostering BDNF expression in RTT. This knowledge, together with a previous publication from the same group demonstrating that HTT modulates BDNF transport, represent the rationale of this novel study. In this manuscript, the authors use genetic and pharmacological approaches to activate HTT, through its selective phosphorylation at S421, thus demonstrating evident amelioration of the Mecp2 null mouse condition.

In general, the technical quality of this work is appropriate, and the whole manuscript is well-written, easy to read and follow and certainly clear also for non-specialists. Conclusions are well supported from data. Although the involvement of BDNF and its modulator HTT are not novel, the presented data and their therapeutic effects on RTT mice are novel. Concerning the medical impact, I have some concerns. Indeed, I believe that too much emphasis has been given so far to BDNF both by the authors in the introduction and generally by the community. As the authors point out, genetic experiments proved that by increasing BDNF levels only some symptoms are ameliorated in mice. In possible good accordance, although clinical trials are still ongoing, first data might indicate only mild improvements. Eventually, in this manuscript the authors use a drug that is actually an aggressive immunosuppressant. Presented data certainly represent a proof of principle of the relevance of S421 in HTT; However, I am not convinced that this molecule might be used in RTT girls.

Eventually, I believe that this work will benefit from some amendments.

In particular:

1. In the abstract the authors suggest the importance of identifying cellular pathways selectively enhancing BDNF signaling in appropriate neuronal circuits. However, in the manuscript they have only analyzed corticostriatal connections. I suggest revising the abstract.
2. Introduction - a) the disorder does not appear only in the second year; b) correct duplications or triplications with duplication or triplication; c) the references Martinowich et al. and Chen et al. 2003 do not prove the involvement of BDNF in the appearance and progression of the disorder; d) I suggest making more clear that rescuing BDNF could not be the definitive solution for RTT.
3. Materials and methods. Although the animal studies make use of quantitative data, in the RTT field is generally recommended to perform blinded studies. However, since the authors do not consider this a pre-clinical study, I believe it is acceptable. What I like much less is the use of homozygous animals for HTT meaning that they never compare siblings. I understand that at this stage there is no solution and that the SA control helps in being confident on the obtained results. The journal should decide whether these approaches are acceptable.
4. All western blots lack MW. Please indicate what are the two PSD95 bands.
5. Supplementary Fig. 3 is pretty bad as it is the case for Fig. 4A. Further, I do not find appropriate to use in the same analysis a different internal standard and run different gels.
6. Fig. 2A is missing all the other genotypes, or at least the WT.
7. Why the authors show the HTT phosphorylation after 2 hrs from FK506 administration while the treatment lasts much longer, and the molecule is administered every 48 hrs? How is HTT phosphorylation when the authors analyze the mouse behavior or at least at the end point of the study?
8. The authors claim that the animals at P30 are symptomatic, but they did not perform any behavioral analysis before treatment (base line of the treated animals); they should at least comment on the gravity of symptoms at this stage.
9. It is not clear to me in the mouse mating procedure, where is the wt male breeder coming from.

Referee #2 (Comments on Novelty/Model System for Author):

Combination of cutting edge in vitro co-culture system in micro fluidic chambers and in vivo Tg mice model.

Referee #2 (Remarks for Author):

In this manuscript the authors demonstrate that promoting HTT phosphorylation at Ser421, restore endogenous BDNF axonal transport in vitro, increased striatal BDNF availability and synaptic connectivity in vivo and improved the phenotype and survival of Mecp2 KO mice, a Rett syndrome model.

This impressive work strengthens and provide new mechanistic insights to previous studies of the same group which demonstrated HTT downregulation in the Mecp2 mice that can be rescued by BDNF.

Overall the data here is convincing, well controlled, the story is in line with their previous results, fit to the aims of EMBO molecular Medicine, and can open new approach for RTT treatments.

Comments:

1) It will be interesting to know whether Mecp2-deficient mice exhibit lower HTT phosphorylation compare to WT. Also showing HTT phosphorylation state in human patient samples, will be very relevant.

2) It will be beneficial and stronger to show some histology for cell death/survival/synapse in addition to the behavioral tests of the treated mice.

Minor Comments:

1) Figure 1: Please add validation for Phospho/non phospho HTT (WB/seq)

2) Figure 2: Add WT/KO WT/KO HTTsd/KO HTTsa in all figures

3) Figure 3: add the WB control for FK506 proper function (from figure 4 to here).

4) All supplementary figures except sup figure 1 are missing titles.

Referee #3 (Comments on Novelty/Model System for Author):

Different experimental replicates have been carried out and the numbers and statistically significant included in the figure legends. Some suggestions for some improvement are included in the comments below.

The authors use both neuronal cultures and a mouse model depleted for the protein which constitutes the highest association to Rett syndrome to date.

This high translational study provides proof that a chemical treatment (easy delivery) has potential therapeutic application for Rett syndrome and other disorders caused by BDNF bioavailability deficits.

Referee #3 (Remarks for Author):

The manuscript "Huntington phosphorylation governs homeostasis and improves the phenotype of Mecp2 knockout mice", by Ehinger et al. proposes a novel strategy to correct or improve disease symptoms in Rett syndrome by the phosphorylation of the Huntingtin protein at the ser 421 in Mecp2 knockout cells and mice, a model for Rett syndrome. The authors show that a Mecp2 knockdown of around 60% causes drastic deficiencies in the transport of BDNF in neuronal cell cultures by reducing both the speed and flow of BDNF vesicles along the axon. BDNF transport defects are prevented by constitutive phosphorylation of HTT at the residue 421 (HTTSD) phosphorylation in vitro and in vivo.

The double mutant mice (Mecp2 ko/ HTTSD) display improved BDNF transport, comparable to that observed in WT mice, as well as increased body weight, increased life span within a given period, reduced apnea, improved motor function and increased night activity. By testing phosphor-TrkB and PSD95 levels the authors report that BDNF restored transport translates in a functional restoration of both BDNF pathway and corticostriatal synapse homeostasis,

The chemical treatment with FK506 (which inhibits calcineurin, responsible to inhibit HTT phosphorylation) had the same BDNF restored transport in culture cells, and phenotype

improvement in the double transgenic mice. The treatment didn't induce any detrimental effect in control mice.

The authors present an interesting and well conducted study supporting that HTT phosphorylation at residue 421 increases the bioavailability of BDNF at the synapse and therefore constitute a directed, rational strategy to consider disorders, like Rett syndrome, where BDNF bioavailability is impaired.

To strengthen the reached conclusions reached and facilitate the reader's understanding, some additional information and modifications should be provided.

Major comments:

- Figure 1. The authors show that HTT phosphorylation at S421 (HTTSD) rescues the transport of BDNF vesicles in siMecp2 neurons. Given the potential interest in therapeutic applications of this strategy, a control of cell viability and homeostasis in this experimental condition is needed.
- Better, cleaner Western blot examples of phospho-TR3KB should be provided in Suppl. Fig. 3. Total TR3KB levels should also be shown (and text modified if necessary).
- Survival curve in Fig. 2B show a time-limited increase in cell survival through a specific window. At the end of the experiment, after day ~90. The authors should provide phosphorylation status of HTT at that later phase of the experiment, or any other evidence that might explain the transient nature of the increased survival effect. The text should modify accordingly to reflect this non-lasting effect.
- Fig 4 support. Show Mecp2 levels in Mecp2 ko mice treated with FK506.
- Are electrophysiological deficits restored in Mecp2 mice after FK506 treatment?

Other comments:

- HTT levels in Suppl. Fig. 1. The Western blot results should be quantified, SD or SEM values calculated, and the number of animals and trials indicated in the figure legend. The text should be updated to reflect the quantification results if necessary.
- Fig. 3 (and text) should reflect the time (after drug treatment) that the shown results are reached. Levels of HTT and HTT phospho Ser421 in treated cells need to be provided.

Minor comments:

- Page 10: correction of the statement, "Cortical neurons were transduced with mCherry-tagged BDNF lentivirus (BDNF-mCherry) and we used using spinning disk confocal videomicroscopy to record the dynamics of BDNF-mCherry containing vesicles within microchannels in the axons (Fig. 1A and Supplement Movie 1)".
- Page 10: Supplementary Fig. 1A is mislabeled as Suppl Fig. 2A in the figures section. Same applies for Suppl. Fig 1B (in text) and Suppl. Fig 2B (in figures section). Please label figures correctly for consistency and the easier read of the paper (first figures to mention first).
- Page 11. Supplementary Fig. 1 should be labeled as "2" since it appears in the manuscript second.
- Page 13. Supplementary Fig 2G should be labeled as 4G.

1st Revision - authors' response

18 October 2019

Reviewer 1

Rett syndrome (RTT) is the first cause of severe intellectual disability in girls and it is generally caused by mutations in MECP2. Although studies on mouse models of this disorder prove that RTT is reversible, at least in mice, no effective cure is yet available. Mecp2 deficiency leads to the deregulation of several genes; among these the down regulation of BDNF is generally accepted as having a role in the disorder. Accordingly, several pre-clinical and clinical trials aim at fostering

BDNF expression in RTT. This knowledge, together with a previous publication from the same group demonstrating that HTT modulates BDNF transport, represent the rationale of this novel study. In this manuscript, the authors use genetic and pharmacological approaches to activate HTT, through its selective phosphorylation at S421, thus demonstrating evident amelioration of the Mecp2 null mouse condition.

In general, the technical quality of this work is appropriate, and the whole manuscript is well-written, easy to read and follow and certainly clear also for non-specialists. Conclusions are well supported from data. Although the involvement of BDNF and its modulator HTT are not novel, the presented data and their therapeutic effects on RTT mice are novel.

Concerning the medical impact, I have some concerns. Indeed, I believe that too much emphasis has been given so far to BDNF both by the authors in the introduction and generally by the community. As the authors point out, genetic experiments proved that by increasing BDNF levels only some symptoms are ameliorated in mice. In possible good accordance, although clinical trials are still ongoing, first data might indicate only mild improvements. Eventually, in this manuscript the authors use a drug that is actually an aggressive immunosuppressant. Presented data certainly represent a proof of principle of the relevance of S421 in HTT; However, I am not convinced that this molecule might be used in RTT girls.

We thank the reviewer for their insightful summary, their appreciation of the novelty of the study, and their open-mindedness toward the work given their very reasonable concerns about its medical impact. We view this as a proof-of-concept study and hope it lays an important foundation for future work.

Eventually, I believe that this work will benefit from some amendments.

1. In the abstract the authors suggest the importance of identifying cellular pathways selectively enhancing BDNF signaling in appropriate neuronal circuits. However, in the manuscript they have only analyzed corticostriatal connections. I suggest revising the abstract.

We have modified the Abstract accordingly.

2. Introduction - a) the disorder does not appear only in the second year; b) correct duplications or triplications with duplication or triplication; c) the references Martinowich et al. and Chen et al. 2003 do not prove the involvement of BDNF in the appearance and progression of the disorder; d) I suggest making more clear that rescuing BDNF could not be the definitive solution for RTT.

We have modified the introduction as suggested.

3. Materials and methods. Although the animal studies make use of quantitative data, in the RTT field is generally recommended to perform blinded studies. However, since the authors do not consider this a pre-clinical study, I believe it is acceptable. What I like much less is the use of homozygous animals for HTT meaning that they never compare siblings. I understand that at this stage there is no solution and that the SA control helps in being confident on the obtained results. The journal should decide whether these approaches are acceptable.

We agree that littermate controls are essential for any behavioral studies comparing genotypes or treatments. We want to point out, however, the tests concerning the HTT phenotype and FK506 treatment were indeed performed blinded. We have clarified this in the material and methods section, page 6.

4. All western blots lack MW. Please indicate what are the two PSD95 bands.

Molecular weights have been added. In addition to the band at 95 kDa protein, western blotting for PSD-95 identifies a lower band (~70 kDa). This lower molecular weight band appears to be a truncated form of PSD-95 lacking the N terminus, as it has been previously noted in several other studies (Topinka & Bredt, 1998; Schlüter et al, 2006; Xu et al, 2008). In this study, we used an antibody against the C terminal part, therefore revealing the truncated form of PSD-95.

5. Supplementary Fig. 3 is pretty bad as it is the case for Fig.4A. Further, I do not find appropriate to use in the same analysis a different internal standard and run different gels.

We have performed new western blots using the same controls, shown now in a new supplementary Figure 3.

6. *Fig. 2A is missing all the other genotypes, or at least the WT.*

All genotypes are now presented in new Figure 2A and as well as in new supplementary Figure 2. The text was modified to mention these results (see page 13).

7. *Why the authors show the HTT phosphorylation after 2 hrs from FK506 administration while the treatment lasts much longer, and the molecule is administered every 48 hrs? How is HTT phosphorylation when the authors analyze the mouse behavior or at least at the end point of the study?*

We thank the reviewer for raising this point. We analyzed phosphorylation levels after 20 days of injections with FK506. This is exactly the same experimental condition as the one used for behavioral analyses. However, at this time point we did not detect elevated HTT phosphorylation in the samples treated with FK506 relative to DMSO-treated animals, likely because repeated treatments desensitize the pathway. Nevertheless, the key observation remains: chronic treatment of FK506 induces neuroprotection in animals and this effect is lost when HTT cannot be phosphorylated at S421 (see new Figure 4).

We kept the western blot showing positive effects of FK506 after acute treatment in Mecp2 KO mice (new Figure 4A) and have moved the acute treatment of WT and HTT_{SA} mice into a new supplementary Figure 3A. We now provide the effect of FK506 chronic treatment as a new Supplementary Figure 3B.

8. *The authors claim that the animals at P30 are symptomatic, but they did not perform any behavioral analysis before treatment (base line of the treated animals); they should at least comment on the gravity of symptoms at this stage.*

This is a valuable suggestion. We have added the following text to the results section page 16: « We choose P30 as a starting point, since the onset of the Mecp2-knockout phenotype is already apparent at this stage as breathing abnormalities, locomotor deficits and body weight loss (Guy *et al*, 2001; Viemari *et al*, 2005; Matagne *et al*, 2017)».

9. *It is not clear to me in the mouse mating procedure, where is the wt male breeder coming from.*

We modified the sentence in the Material and Methods section page 5 to clarify: “The Mecp2^{tm1-1Bird} Mecp2-deficient mice were obtained from the Jackson Laboratory and maintained on a C57BL/6 background by using C56BL/6J male breeders (also from the Jackson Laboratory).”

Reviewer 2

In this manuscript the authors demonstrate that promoting HTT phosphorylation at Ser421, restore endogenous BDNF axonal transport in vitro, increased striatal BDNF availability and synaptic connectivity in vivo and improved the phenotype and survival of Mecp2 KO mice, a Rett syndrome model.

This impressive work strengthens and provide new mechanistic insights to previous studies of the same group which demonstrated HTT downregulation in the Mecp2 mice that can be rescued by BDNF.

Overall the data here is convincing, well controlled, the story is in line with their previous results, fit to the aims of EMBO molecular Medicine, and can open new approach for RTT treatments.

We thank the reviewer for their appreciation of the work.

1) *It will be interesting to know whether Mecp2-deficient mice exhibit lower HTT phosphorylation compare to WT. Also showing HTT phosphorylation state in human patient samples, will be very relevant.*

We now provide a new supplementary Figure 3B, a western blot of HTT phosphorylation levels in WT and Mecp2 KO mice showing that there is no difference between the two genotypes.

RTT human samples would be highly relevant to this study but, unsurprisingly, are very difficult to obtain. Even so, our experience in using human samples from HD patients has shown that variability of post-mortem preparations makes it extremely difficult to obtain reliable signals for HTT phosphorylation.

2) *It will be beneficial and stronger to show some histology for cell death/survival/synapse in addition to the behavioral tests of the treated mice.*

As suggested by the reviewer, we now provide a full new figure (Supplementary Figure 4) in which we analyzed the presence of cleaved-caspase-3 and TUNEL in Mecp2 KO mice and the effect of FK506 treatment. We did not detect any cell death in Mecp2 KO mice nor any increase in cell death after 20 days of chronic treatment. We now report on page 16: “We next investigated cell death in the striatum of these mice. In accordance with previous reports (Armstrong, 2005; Kishi & Macklis, 2004; Reiss *et al*, 1993; Roux *et al*, 2007), we did not detect any cell death in Mecp2 knockout mice. Importantly, FK506 treatment did not induce any neuronal toxicity or cell death after 20 days of treatment (Supplementary Fig. 4A and B).”

We also investigated cFos as a marker of neuronal activity. We observed a trend toward reduced cFos staining in Mecp2 knockout that was restored to control levels in treated mice. We present these results in the new Supplementary Fig. 4C and D and on page 16.

Minor Comments:

1) *Figure 1: Please add validation for Phospho/non phospho HTT (WB/seq)*

We previously reported that S to A mutation at S421 leads to loss of HTT phosphorylation as revealed by the use of a phospho-S421 antibody (Colin *et al*, 2008). We provide evidence in the new supplementary Figure 1C and supplementary figure 3A that the S421A mutation abrogates recognition of phosphorylated HTT by the phospho-specific HTT antibody. This validation of the phospho/non phospho HTT is provided in the text (page 12).

2) *Figure 2: Add WT/KO WT/KO HTTsd/KO HTTsa in all figures*

We now provide all the different genotypes in new Figure 2 and in new supplementary Figure 1G. The text was modified accordingly (page 13).

3) *Figure 3: add the WB control for FK506 proper function (from figure 4 to here).*

The western blotting analysis of previous Figure 4A showing FK506 effect on WT mice was moved to supplementary Figure 3A as requested by Reviewer 1, as these experiments were performed in mice and not in neuronal cultures.

4) *All supplementary figures except sup figure 1 are missing titles.*

We apologize for this. Titles have been added to all the supplementary figures.

Reviewer 3

The manuscript "Huntington phosphorylation governs homeostasis and improves the phenotype of Mecp2 knockout mice", by Ehinger et al. proposes a novel strategy to correct or improve disease symptoms in Rett syndrome by the phosphorylation of the Huntingtin protein at the ser 421 in Mecp2 knockout cells and mice, a model for Rett syndrome. The authors show that a Mecp2 knockdown of around 60% causes drastic deficiencies in the transport of BDNF in neuronal cell cultures by reducing both the speed and flow of BDNF vesicles along the axon. BDNF transport defects are prevented by constitutive phosphorylation of HTT at the residue 421 (HTTSD) phosphorylation in vitro and in vivo.

The double mutant mice (Mecp2 ko/ HTTSD) display improved BDNF transport, comparable to that observed in WT mice, as well as increased body weight, increased life span within a given period, reduced apnea, improved motor function and increased night activity. By testing phosphor-TrkB and PSD95 levels the authors report that BDNF restored transport translates in a functional restoration of both BDNF pathway and corticostriatal synapse homeostasis,

The chemical treatment with FK506 (which inhibits calcineurin, responsible to inhibit HTT phosphorylation) had the same BDNF restored transport in culture cells, and phenotype

improvement in the double transgenic mice. The treatment didn't induce any detrimental effect in control mice.

The authors present an interesting and well conducted study supporting that HTT phosphorylation at residue 421 increases the bioavailability of BDNF at the synapse and therefore constitute a directed, rational strategy to consider disorders, like Rett syndrome, where BDNF bioavailability is impaired.

We appreciate the reviewer's interest in this work.

To strengthen the conclusions reached and facilitate the reader's understanding, some additional information and modifications should be provided.

Major comments:

- Figure 1. The authors show that HTT phosphorylation at S421 (HTT_{SD}) rescues the transport of BDNF vesicles in siMecp2 neurons. Given the potential interest in therapeutical applications of this strategy, a control of cell viability and homeostasis in this experimental condition is needed.

We agree with the importance of demonstrating the lack of toxicity in our transport assays. Therefore, we performed cell viability assays using MTT. Tests were performed on neurons from different genotypes cultured in the same condition as the transport assays shown in Figure 1. We found that neither HTT genotype (WT, HTT_{SD} or HTT_{SA}) nor Mecp2 silencing had any effect on cell viability. The effects on transport are therefore not due to differences in cell viability. See new Figure 1C and F.

- Better, cleaner Western blot examples of phospho-TrkB should be provided in Suppl. Fig. 3. Total TrkB levels should also be shown (and text modified if necessary).

We have improved the quality of the western blots that are now shown in the new supplementary Figure 1G.

Total TrkB would have been interesting to quantify. Unfortunately, microdissection of the cortex and striatum in our mice (KO, WT, KO/HTT_{SA} and KO/HTT_{SD}) collects less than 100 micrograms of tissue. With the samples obtained we were able to quantify Phospho TrkB, PSD95 and GAPDH, but we no longer have enough to perform the total TrkB assay.

- Survival curve in Fig. 2B show a time-limited increase in cell survival through a specific window. At the end of the experiment, after day ~90. The authors should provide phosphorylation status of HTT at that later phase of the experiment, or any other evidence that might explain the transient nature of the increased survival effect. The text should modify accordingly to reflect this non-lasting effect.

We believe the reviewer is referring to the limited increase in mouse, not cell, survival. The specific window of survival mentioned by the reviewer likely reflects a shift in the survival of the KO HTT_{SD} mice compared to the KO mice. Indeed, although the HTT_{SD} mutation ameliorates behavior and extends the lifespan of the mice, the mutation is only protective until a certain time point. Importantly, the KO HTT_{SD} survival curve never crosses the KO survival curve and the effect is statistically significant. The text has been modified in the results section (page 13). It is important to note that FK506 treatment was not transient and significantly prolonged lifespan (Figure 4B).

The reviewer is also requesting an analysis of HTT phosphorylation status. However, given the nature of the mutation, (S into A or S into D), the phospho-epitope on S421 is absent in these mice (new supplementary Figure 1C).

- Fig 4 support. Show Mecp2 levels in Mecp2 ko mice treated with FK506.

We are not sure we understand the reviewers' concern: Mecp2 protein cannot be detected in Mecp2 KO mice (Guy et al., 2001). If the reviewer is asking for Mecp2 protein levels in WT mice with FK treatment, we provide this information in the new supplementary Figure 3C. Briefly, there are no differences in Mecp2 levels between treatments.

-Are electrophysiological deficits restored in Mecp2 mice after FK506 treatment?

This is an interesting study that we intend to pursue in the future, but we believe that this is beyond the scope of the current manuscript.

Other comments:

- *HTT levels in Suppl. Fig. 1. The Western blot results should be quantified, SD or SEM values calculated, and the number of animals and trials indicated in the figure legend. The text should be updated to reflect the quantification results if necessary.*

The western blot of old supplementary Figure 1 has been performed again and was quantified. It is now shown in the new supplementary Figure 1C and the text (page 12) and legend modified accordingly.

- *Fig. 3 (and text) should reflect the time (after drug treatment) that the shown results are reached. Levels of HTT and HTT phosphor Ser421 in treated cells need to be provided.*

The western blot of the previous supplementary Figure 5 showing levels of HTT and HTT phosphorylation has been moved to the new Figure 3A, with quantification.

Minor comments:

- *Page 10: correction of the statement, "Cortical neurons were transduced with mCherry-tagged BDNF lentivirus (BDNF-mCherry) and we used using spinning disk confocal videomicroscopy to record the dynamics of BDNF-mCherry containing vesicles within microchannels in the axons (Fig. 1A and Supplement Movie 1)".*

- *Page 10: Supplementary Fig. 1A is mislabeled as Suppl Fig. 2A in the figures section. Same applies for Suppl. Fig 1B (in text) and Suppl. Fig 2B (in figures section). Please label figures correctly for consistency and the easier read of the paper (first figures to mention first).*

- *Page 11. Supplementary Fig. 1 should be labeled as "2" since it appears in the manuscript second.*

- *Page 13. Supplementary Fig 2G should be labeled as 4G.*

We thank the reviewer for pointing out these typos, which we have now corrected.

Literature cited

- Armstrong DD (2005) Neuropathology of Rett syndrome. *J. Child Neurol.* **20**: 747–753
- Colin E, Zala D, Liot G, Rangone H, Borrell-Pagès M, Li X-J, Saudou F & Humbert S (2008) Huntingtin phosphorylation acts as a molecular switch for anterograde/retrograde transport in neurons. *EMBO J.* **27**: 2124–2134
- Guy J, Hendrich B, Holmes M, Martin JE & Bird A (2001) A mouse Mecp2-null mutation causes neurological symptoms that mimic Rett syndrome. *Nat. Genet.* **27**: 322–326
- Kishi N & Macklis JD (2004) MECP2 is progressively expressed in post-migratory neurons and is involved in neuronal maturation rather than cell fate decisions. *Mol. Cell. Neurosci.* **27**: 306–321
- Matagne V, Ehinger Y, Saidi L, Borges-Correia A, Barkats M, Bartoli M, Villard L & Roux J-C (2017) A codon-optimized Mecp2 transgene corrects breathing deficits and improves survival in a mouse model of Rett syndrome. *Neurobiol. Dis.* **99**: 1–11
- Reiss AL, Faruque F, Naidu S, Abrams M, Beaty T, Bryan RN & Moser H (1993) Neuroanatomy of Rett syndrome: a volumetric imaging study. *Ann. Neurol.* **34**: 227–234
- Roux J-C, Dura E, Moncla A, Mancini J & Villard L (2007) Treatment with desipramine improves breathing and survival in a mouse model for Rett syndrome. *Eur. J. Neurosci.* **25**: 1915–1922
- Schlüter OM, Xu W & Malenka RC (2006) Alternative N-terminal domains of PSD-95 and SAP97 govern activity-dependent regulation of synaptic AMPA receptor function. *Neuron* **51**: 99–111
- Topinka JR & Brecht DS (1998) N-terminal palmitoylation of PSD-95 regulates association with cell membranes and interaction with K⁺ channel Kv1.4. *Neuron* **20**: 125–134
- Viemari J-C, Roux J-C, Tryba AK, Saywell V, Burnet H, Peña F, Zanella S, Bévençut M, Barthelemy-Requin M, Herzing LBK, Moncla A, Mancini J, Ramirez J-M, Villard L & Hilaire G (2005) Mecp2 deficiency disrupts norepinephrine and respiratory systems in mice. *J. Neurosci.* **25**: 11521–11530

Xu W, Schlüter OM, Steiner P, Czervionke BL, Sabatini B & Malenka RC (2008) Molecular dissociation of the role of PSD-95 in regulating synaptic strength and LTD. *Neuron* **57**: 248–262

2nd Editorial Decision

14 November 2019

Thank you for the submission of your revised manuscript to EMBO Molecular Medicine. We have now received the enclosed reports from the referees that were asked to re-assess it. As you will see the reviewers are now supportive and I am pleased to inform you that we will be able to accept your manuscript pending minor editorial amendments.

I look forward to receiving a new revised version of your manuscript within 2 weeks.

***** Reviewer's comments *****

Referee #1: Suitable for publication

Referee #2: Suitable for publication

2nd Revision - authors' response

27 November 2019

Authors made the requested changes.

Corresponding Author Name: Jean-Christophe ROUX and Frederic Saudou

Journal Submitted to: EMBO MOLECULAR MEDICINE

Manuscript Number: EMM-2019-10889